# Climate: The dominant factor influencing the spatial distribution pattern of the leaf trait network of *Populus euphratica* along the main stream of the Tarim River

ChengZhi Peng[1][◎], ShiYu Yao[1][◎], WenJuan Huang[1,2]*, Jie Wang[1], ShuangFei Song[1], Pei Zhang[1], Peipei Jiao[1,2]

**1** College of Life Science and Technology, Tarim University, Alar, Xinjiang Province, P. R. China, **2** Xinjiang Production & Construction Corps Key Laboratory of Protection and Utilization of Biological Resources in Tarim Basin, Tarim University, Alar, Xinjiang Province, P. R. China

◎ These authors contributed equally to this work.
* hwjzky@163.com

## Abstract

Leaves are the primary interface through which plants interact with the environment, their functional traits (morphology, anatomy, physiology) directly reflecting ecological strategies that mediate species-environment interactions. These traits link plant performance to ecosystem processes, shaping species distributions and coexistence via their complex relationships with climatic and edaphic factors. Based on previous work, we selected 20 *P. euphratica* trees along the desert riparian forest of the main stream of the Tarim River for leaf sample collection and habitat survey. We used 27 leaf traits of *P. euphratica* to visualize the leaf trait network (LTN). Through network structure parameters, such as edge density, diameter, average path length, and average clustering coefficient, the spatial pattern of the LTN and its relationships with 19 climatic factors and 11 soil factors were discussed using principal component analysis and correlation analysis. The results showed that: (1) there were significant differences in the parameters of the leaf trait network of *P. euphratica* along the main stream of the Tarim River. The variation coefficients of the diameter and average path length were the largest, respectively, whereas that of the average clustering coefficient was the smallest. (2) Among the parameters, only Modularity was significantly correlated with STK and SOM. But Average clustering coefficient was significantly positively correlated with Isothermality, Average path length and Diameter were significantly positively correlated with Min temperature of coldest month and the Average clustering coefficient was significantly negatively correlated with Min temperature of coldest month, the Diameter was significantly positively correlated with Precipitation of wettest month. In general, the correlation between climate factors and LTNs was stronger than soil factors. (3) The explanatory power of climatic factors alone on the leaf traits of *P. euphratica* was generally higher than that of soil factors, indicating

**Data availability statement:** The data that supports the findings of this study are available in the supplementary material of this article. The data that support the findings of this study are openly available in the WorldClim database at https://www.worldclim.org.

**Funding:** The funding agency of the National Natural Science Foundation of China(Grant number 31160110) had no role in study design, data collection and analysis, decision to publish, or preparation of the manuscript.

**Competing interests:** The authors have declared that no competing interests exist.

that climatic conditions play a more decisive role in shaping the network structure of leaf traits of *P. euphratica*. However, the influence of soil conditions on some LTNs parameters cannot be ignored.The spatial variability of leaf trait networks is driven by climate and soil factors, with climate dominating along the Tarim River's main course.

## Introduction

Plant functional traits refer to the morphological, physiological, biochemical, and behavioral characteristics of plants in ecosystems. These characteristics are closely related to ecosystem functions and reflect the strategies used by plants to cope with environmental changes [1,2]. As an important aspect of plant functional traits, leaf traits are not only closely related to plant growth, development, and reproduction but are also affected by the external environment and phylogenetic development. It is a bridge between plants and the environment, showing the adaptability and self-regulation ability of plants to complex habitats [3–5]. Under environmental stress, plants usually improve their adaptability by adjusting their leaf functional traits. In addition, as the main organ of photosynthesis and material production, leaves are key to material exchange and energy conversion in plants [6]. Leaf traits directly reflect the photosynthetic capacity of plants and their strategies of resource acquisition, utilization, and distribution [7,8]. Therefore, it is important to understand the changes in leaf functional traits in different environments to explore the environmental adaptability of plants. This helps to reveal the ecological adaptation mechanisms of plants and provides an important scientific basis for the maintenance and management of ecosystem functions.

With the intensification of global environmental change, research on variations in leaf functional traits and their adaptation mechanisms has attracted attention. Studying the changes in plant functional traits with environmental factors is helpful for understanding the physiological and ecological mechanisms of plants under climate change conditions and is of great significance for understanding the survival and distribution of plants [9]. Additionally, climate change and other environmental factors have significant effects on leaf functional traits [10–16]. These variations result from the interaction of environmental factors and phylogenetic history, showing dynamic changes in global, regional, and local distributions [17–19]. The relationship between traits and the environment reflects the optimal adaptation principles for plant growth and adaptation under natural conditions [20]. Therefore, leaf functional traits provide information on environmental changes as well as directly reflecting the survival strategies of plants to adapt to environmental changes.

The economic spectrum of plant functional traits reveals an important relationship among plant functional traits. Plant leaf traits are not isolated but are closely related to many other traits [21,22]. Recently, researchers have proposed various theories and methods regarding plant trait networks (PTNs) [23]. By quantifying the complex relationships between multiple leaf traits, a multidimensional network was

constructed to explain the relationships between different leaf traits. The study of plant trait networks can comprehensively, multi-dimensionally, and visually analyze and evaluate the relationship between traits, network topology, and hub traits [24]. This allows researchers to explore the key traits of plants in different environments and their relationships with the environment and provide a new way to reveal the adaptation and response mechanisms of plants to environmental and resource changes. The purpose of this study was to explore how the leaf trait network of *P. euphratica* along the main stream of the Tarim River responds to environmental change. By analyzing the spatial variability and driving factors, the adaptive strategies of *P. euphratica* leaf traits in its environment are revealed.

The environment of the main stream of the Tarim River in Xinjiang is complex and varied. It is a typical ecologically fragile area with strong environmental sensitivity, water shortages, sparse vegetation, and severe soil and wind erosion. There are vast desert riparian forests on both sides of the river, and the vegetation is dominated by *P. euphratica*. It is an important tree species in the Salicaceae family. It has strong stress resistance and can adapt to harsh environments such as drought, wind erosion, and saline-alkali soil [25]. It plays an active role in regulating the climate, preventing wind and sand erosion, preventing desert expansion, and protecting oases. Understanding the interactions between *P. euphratica* leaves and the environment is of great significance for revealing the mechanisms of plant adaptation to drought-prone environments. In a previous study [26], the relationship between leaf traits and the environment of *P. euphratica* has been studied, this manuscript is the statistical rework of the mentioned study. The influence of the climate drought index and river water flow on the leaf trait network of *P. euphratica* was explored using the plant trait network analysis method by determining leaf morphology and functional traits of *P. euphratica* along the main stream of the Tarim River. The lower stratum corneum (LSC), upper stratum corneum (USC), and midvein vascular bundle (MVB) were the central traits of the leaf trait network of *P. euphratica*. It optimizes resource utilization and improves drought resistance by adjusting its leaf traits, thereby exhibiting a high degree of environmental adaptability. However, there are a few reports on the complex relationship network between the multiple leaf traits of *P. euphratica* and their relationships with the environment [27,28].

Although traditional methods have extensively studied the relationships between single traits or pairs of traits and the environment [10,29], the synergistic interactions among leaf traits may more profoundly reflect plant adaptive mechanisms [24]. For instance, Li et al.[27] highlighted that the topological structure of the leaf trait network (LTN) can capture climate-driven trait modular shifts, whereas direct analysis of single traits may overlook the synergistic responses of multiple traits. This study adopted the LTN approach because it systematically deciphers the complex associations between traits and reveals how environmental factors influence plant functionality by modulating the structure of trait networks.

This study used the *P. euphratica* forest on the desert bank of the main stream of the Tarim River in Xinjiang as the research object, By incorporating multi-dimensional environmental drivers (soil properties and climatic factors) and integrating spatial variability analysis, we employed leaf trait network and principal component analysis (PCA), variance decomposition analysis (VPA), and other analysis methods to construct the *P. euphratica* leaf trait network along the main stream of the Tarim River. The spatial pattern of the leaf trait network in the entire basin and its relationship with environmental factors, such as climate and soil, were analyzed. The purpose of this study was to analyze how *P. euphratica* adapts to the environment of the main stream of the Tarim River through the integration of leaf traits and adjustment of the relationship between traits, to reveal the adaptation strategy of *P. euphratica* to the environment, ultimately protecting wild *P. euphratica* forests along the Tarim River.

## Materials and methods

### Overview of the study area and site information

The Tarim River flows through the northern Tarim Basin in northwestern China. It is the fifth-longest inland river in the world and the longest river in China. Its main stream is formed by the confluence of the Aksu River, which originates in the Tianshan Mountains, and the Yarkand and Hotan Rivers, which originate in the Kunlun Mountains in the city of Aral.

Affected by the climate, the seasonal variation of runoff indicates that the Tarim River is seasonal river [30]. This study was conducted in July 2022, and 20 survey sampling sites were established in the natural *P. euphratica* forest on both sides of the riverbank from east to west along the main stream of the Tarim River, from the source to the end of the river (Fig 1). The sample plots were selected uniformly distant from human disturbance. The latitude and longitude, altitude, vertical riparian distance, and other data for each sample plot were recorded simultaneously. Basic information on the 20 sample plots is provided in S1 Table in S1 File.

### Leaf sample collection and pretreatment

Each plot was measured for each tree, and 15 mature *P. euphratica* trees with good growth status and consistent diameter at breast height (DBH) in the range of 20–30 cm were randomly selected as sample plants and their specific DBH was recorded. High-branched scissors were used for sampling. The sampling site covered approximately one third of the lower crown of each *P. euphratica* tree. Healthy leaves at the penultimate third node of the twig were collected from the east, south, west, and north sections of each tree. After sampling was completed, all leaves of the same plant were evenly mixed, and 20 leaves were randomly selected for cutting but retained the main vein and then placed in a reagent bottle containing FAA fixative (70% ethanol: formaldehyde: acetic acid = 90:0.5:0.5). Another 100 leaves were randomly selected and the petiole was removed. The fresh weight of the leaves was first measured, followed by dispersing the leaves flat on a black background plate placed on a ruler, and photographs of the vertical leaves taken for subsequent analyses of leaf morphology indices (Fig 2a). After processing, the samples were returned to the laboratory for subsequent analyses.

### Soil sample collection and pretreatment

Soil profiles were dug near each plant, and three soil samples were taken from each site using a soil auger at depths of (0,20] cm, (20,40] cm, (40,60] cm, (60,80] cm, and (80,100] cm for fresh soil columns. Part of the soil from the base of the soil columns was put into an aluminum box and transported, weighed, and then dried at 105°C in a thermostatic drying

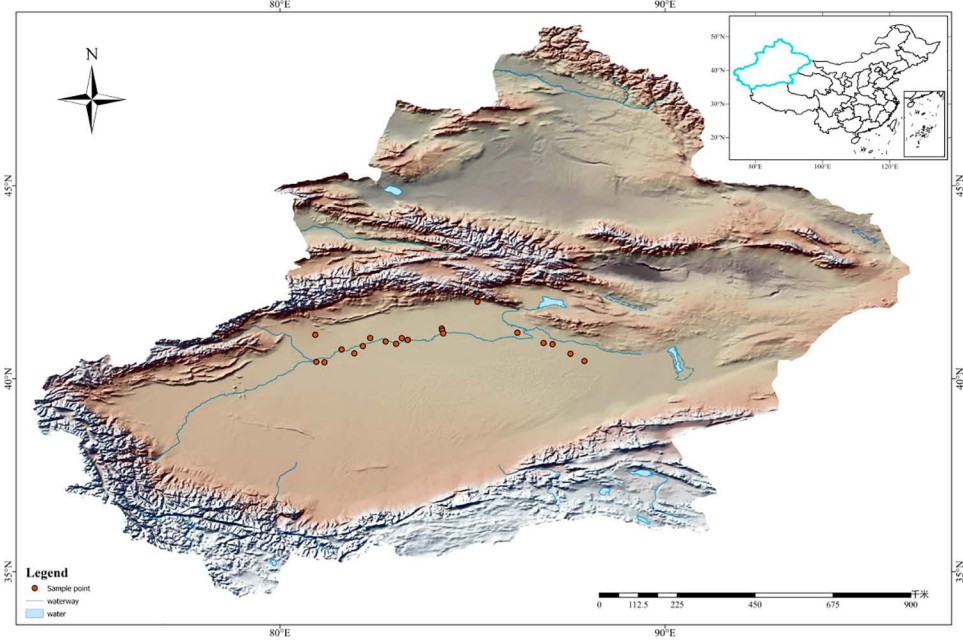

**Fig 1. Map of the sampling points.**

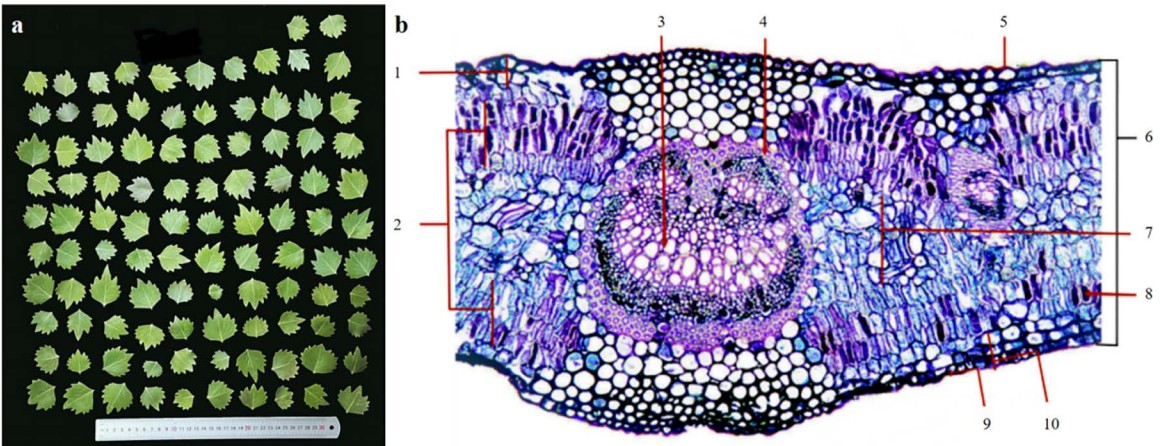

**Fig 2. Leaf morphology and leaf anatomical structure. (a) Leaf morphology. (b) Leaf anatomical structure: (1)** Upper epidermis; **(2)** Palisade tissue; **(3)** Vascular bundle; **(4)** Sclerenchyma; **(5)** Upper stratum corneum; **(6)** Leaf thickness; **(7)** Spongy tissue; **(8)** Mucous cells; **(9)** Lower stratum corneum; **(10)** Lower epidermis.

oven to determine dry weight and the moisture content of the soil. The rest of the soil samples were transported in sealed bags. Roots and impurities were manually removed and dried naturally in a cool, ventilated place. After air-drying, the soil samples were ground and passed through a 2 mm soil sieve. One part was used to determine soil pH (pH), total salt content (TS), and electrical conductivity (EC), and the other part was ground again and passed through a 0.15 mm soil sieve to determine soil total nitrogen (STN), soil total phosphorus (STP), soil total potassium (STK), and soil organic matter (SOM).

## Determination of leaf traits

Leaves collected from the field were washed with deionized water and placed in a constant-temperature drying oven at 105°C for 30 min. The leaves were then dried to a constant weight at 80°C, and their dry weight was measured. The dried leaves were crushed using a pulverizer (15000 r/min), ground, and sieved (0.2 mm) to determine the chemical elements in the leaves. Based on the standard plant trait measurement method [31], we measured 27 leaf traits of *P. euphratica* and divided them into three categories according to their functions: eight leaf morphological traits, including leaf water content (LWC) and specific leaf area (SLA); seven stoichiometric traits, including leaf total nitrogen (LN) and leaf total phosphorus (LP); and 12 anatomical structural traits, including leaf sponge tissue (ST) and leaf palisade tissue (PT). The specific names, abbreviations, and unit information of the traits are detailed in S2 Table in S1 File.

The carbon content of leaf samples was determined using the potassium dichromate oil bath heating method. The nitrogen content was determined using the Kjeldahl method with a Kjeldahl nitrogen analyzer (Hanon, K9840, Shanghai, China). Phosphorus content was determined using an ultraviolet spectrophotometer (INESA, L8, Shanghai, CHINA) following the molybdenum antimony scandium colorimetric method. The potassium content was determined using a flame photometer (INESA, FP4231, Shanghai, CHINA) and the $H_2SO_4$-$H_2O_2$ digestion method. The morphological characteristics of the leaves were analyzed using ImageJ (1.53t, 2022). The anatomical structures of the leaves were observed using the paraffin sectioning method (Fig 2b). The paraffin sections were observed under a digital microscope (Leica, Wetzlar, Germany) and analyzed using an image processor (Leica Application Suite V4.0.0 DVD, Wetzlar, Germany).

## Climate data and soil data acquisition

Data for the 19 climatic factors in this study (S3 Table in S1 File) were obtained from the WorldClim database (www.worldclim.org) [32]. The determination of soil included water content (WC), total salt content (TS), soil organic matter (SOM),

soil total nitrogen (STN), soil total phosphorus (STP), soil total potassium (STK), soil carbon-nitrogen ratio (SC:N), soil carbon-phosphorus ratio (SC:P), soil nitrogen-phosphorus ratio (SN:P), electrical conductivity (EC), and soil pH (S4 Table in S1 File). TS was determined using the residue-drying method. WC was determined using the aluminum box drying method. EC was obtained by measuring the soil leaching solution using a conductivity meter. The pH of the soil leachate was measured using a pH meter. The STN and SOM contents were determined using a FLASHSMART elemental analyzer (Thermo Fisher, FlashSmart CN/CNS, GERMANY). The STP was determined using a sodium hydroxide melting method-ultraviolet spectrophotometer (INESA, L8, Shanghai, CHINA). STK was determined using the sodium hydroxide extraction-flame photometric method (INESA, FP4231, CHINA).

### Leaf trait network parameters and their ecological significance

The leaf trait network (LTNs) is a multidimensional network comprising nodes and edges. Nodes represent leaf traits and edges represent relationships between traits. The strength of the trait relationship was determined by calculating the absolute value of Pearson's correlation coefficient ($|r|$, $r > 0.2$). The correlation coefficient of $P < 0.05$ was retained and set to 1. The correlation coefficient below the threshold was set to zero, and an adjacency matrix A=[ai, j] was generated. The edges were then weighed and visualized [33].

The overall characteristics of LTNs are described by their diameter, average path length, edge density, modularity, and average clustering coefficient. The diameter represents the maximum and minimum distances between any two nodes in the network. The average path length is the average shortest path between all the nodes. Edge density is the ratio of the actual edge to the maximum possible edge. A short diameter, short path, and high edge density indicate high synergy between traits. Modularity describes the degree of separation between subnetworks [34], and high modularity indicates that the functional modules are clear. The average clustering coefficient represents the average of clustering coefficient of each trait in the leaf trait network [35], and a high value indicates that some traits have good synergy. The node parameters of LTNs include degree, closeness, and betweenness, which are used to quantify the relationships between traits and identify their topological roles and adaptability in LTNs [36]. The degree represents the number of edges connected to the node, and the high-degree trait is the hub trait of the network. Closeness is the reciprocal of the shortest path length from a specific node to other nodes. The high-closeness trait is closely related to other traits. Betweenness refers to the number of shortest paths through a node, and the high-betweenness trait is the *bridge* or *intermediary* of the functional module.

### Data analyses

Excel 2007 was used to organize the experimental data, and SPSS19.0 software was used to analyze the data. Network construction and statistical analyses were performed using Origin Drawing (2021) and R (4.3.0 version, 2023).

The R 'igraph' package was used to calculate the parameters of node traits and total LTNs. Variance partitioning analysis (VPA) was performed using the "vegan" package in R to quantify the proportional contributions of climate, soil, and their interaction effects on network parameters. Permutation tests (999 permutations) were conducted via the permutest function to assess the statistical significance of these contributions (*p*-values) at a significance level of α=0.05. This approach was used to interpret the effects of different environmental factors (climate and soil) on the variability of network parameters.

## Results

### Leaf trait variability of *P. euphratica* in the main stream of Tarim River

Based on the leaf trait dataset of 300 *P. euphratica* trees in 20 plots along the main stream of the Tarim River, spatial variability was studied. The individual variation characteristics of 27 traits of *P. euphratica* leaves in the main stream of the Tarim River (Fig 3a) and the overall variation characteristics of three types of traits (leaf morphological, stoichiometric, and anatomical structure traits) (Fig 3b) are shown.

The degree of variation in the leaf morphological traits ranged from 7.60 to 29.86%. The degree of variation in leaf WC (7.60%) was the smallest, showing weak variation, and the remaining leaf morphological traits showed medium variation. Among them, the single-leaf area (29.87%) and dry weight of a single leaf (29.69%) showed the highest variability.

The degree of variation in the leaf stoichiometric characteristics was between 8.50 and 75.79%. Only the degree of variation of LC (8.50%) was the smallest, showing weak variation. The remaining leaf stoichiometric characteristics showed medium variation, among which the variation in the leaf C:P ratio (5.79%) was the highest.

The degree of leaf anatomical structure trait variation ranged from 9.98 to 47.27%; only the degree of variation of the cell tension ratio (9.98%) was the smallest, and the other leaf anatomical structure traits showed medium variation, among which the variation in the midrib vascular bundle (47.27%) was the largest.

The overall leaf anatomical traits of the main stream of the Tarim River showed 28.68% variation, the overall leaf morphological traits showed 20.09% variation, and the overall leaf stoichiometric traits showed 42.02% variation. Leaf stoichiometry exhibited the largest variation among the three leaf traits.

### Spatial variability of leaf trait network of *P. euphratica* in the main stream of Tarim River

The network of each plot and whole-leaf traits was constructed (Fig 4), and the network parameters were calculated. In the entire reach of the main stream of the Tarim River, the maximum, minimum, and average values of the network parameters of the 20 sample points (S5 Table in S1 File) and the variability of the overall network parameters were as follows (Fig 5): the maximum average path length of the network was 4.60, the minimum value was 1.34, the average value was 2.15, and the coefficient of variation was 32.97%. The maximum diameter was 11.17, the minimum diameter was 3.09, the average diameter was 5.17, and the coefficient of variation was 35.36%. The maximum edge density was 0.25, the minimum was 0.10, the average was 0.14, and the coefficient of variation was 25.18%. The maximum value of the average clustering coefficient was 0.73, the minimum value was 0.43, the average value was 0.54, and the coefficient of variation was 17.21%. The maximum modularity was 0.67, the minimum was 0.27, the average was 0.56, and the coefficient of

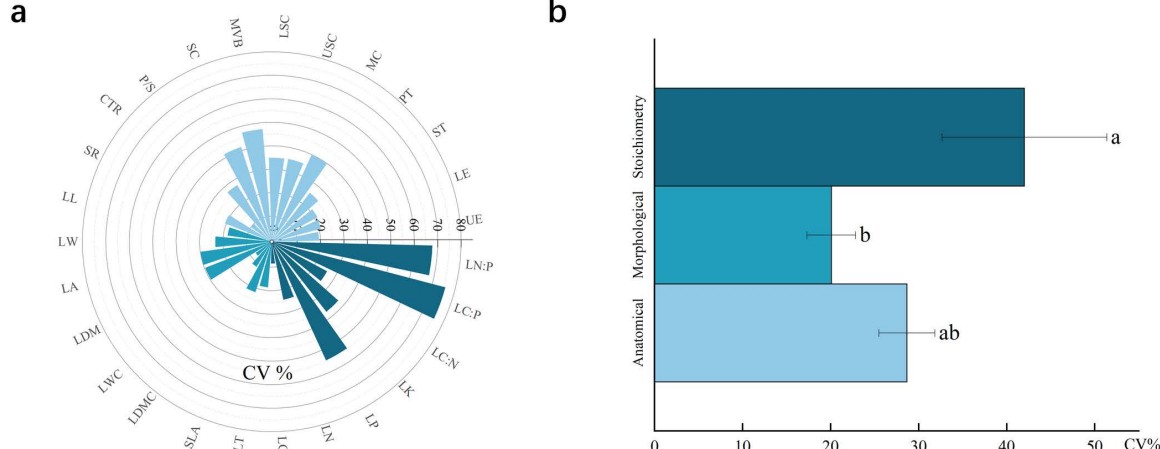

**Fig 3. The coefficient of variation of 27 leaf traits and the overall variation characteristics of the three types of leaf traits. (a) The coefficient of variation for the 27 leaf traits. (b) The overall variation characteristics of three types of leaf traits.** (CTR: Cell tension ratio, P/S: The ratio of palisade tissue to sponge tissue, SC: Sclerenchyma, MVB: Median vascular bundle, LSC: Lower stratum corneum, USC: Upper stratum corneum, MC: Mucous cells, PT: palisade tissue, ST: Spongy tissue, LE: Lower epidermal thickness, UE: Upper epidermal thickness, LN:P: Leaf nitrogen to phosphorus ratio, LC:P: Leaf organic matter to phosphorus ratio, LC:N: Leaf organic matter to nitrogen ratio, LK: Leaf total potassium, LP: Leaf total phosphorus, LN: Leaf total nitrogen, LC: Leaf total organic matter, LT: Leaf thickness, SLA: Specific leaf area, LDMC: Leaf dry matter content, LWC: Leaf water content, LDW: Leaf dry weight, LA: Leaf area, LW: Leaf width, LL: Leaf length, and SR: Spongy ratio).

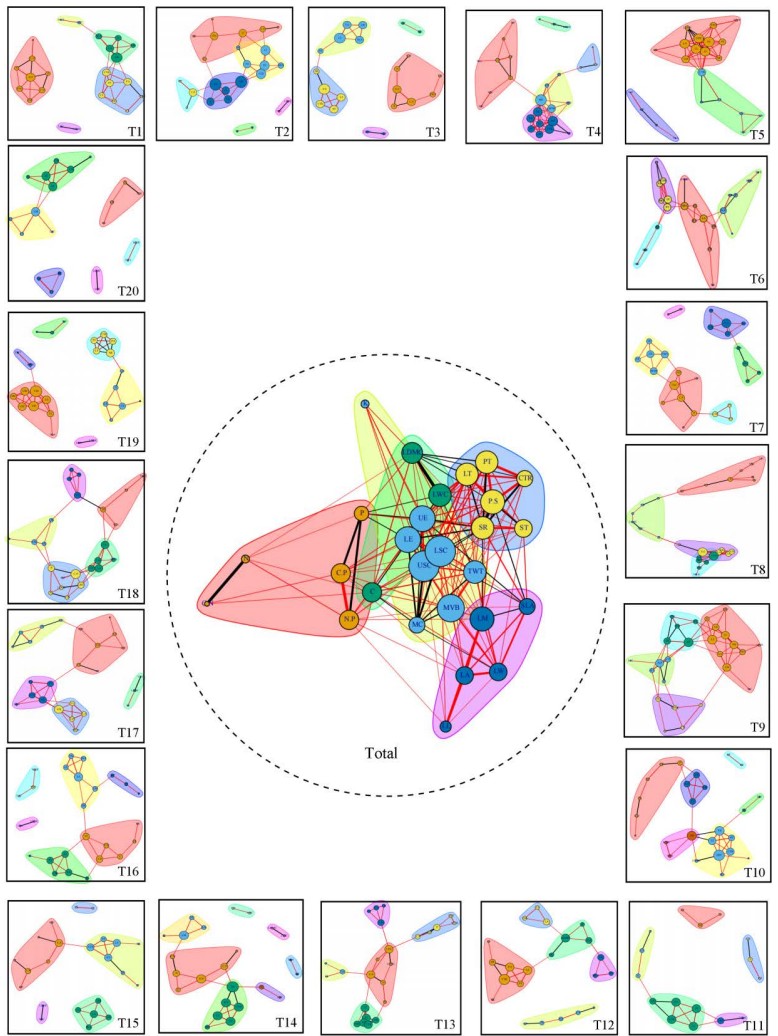

**Fig 4. Leaf trait network and whole leaf trait network of _P. euphratica_ in 20 sampling sites (T1–T20) in the main stream of Tarim River.** Note: Features with the same background color belong to the same module; the red line shows a positive correlation, the black line shows a negative correlation, the line width indicates the strength between traits, and the node size indicates the degree of the trait. K: Leaf total potassium, LDMC: Leaf dry matter content, PT: Palisade tissue, LT: Leaf thickness, CTR: Cell tension ratio, LWC: Leaf water content, P.S: The ratio of palisade tissue to sponge tissue, P: Leaf total phosphorus, UE: Upper epidermis thickness, SR: Spongy ratio, ST: Sponge tissue, LE: Lower epidermis thickness, LSC: Lower stratum corneum, USC: Upper stratum corneum, TWT: Sclerenchyma, N: Leaf total nitrogen, C.P: Leaf organic matter to phosphorus ratio, C: Leaf total organic matter, MVB: Midvein vascular bundle, LM: Leaf dry weight, SLA: Specific leaf area, C.N: Leaf organic matter to nitrogen ratio, N.P: Leaf nitrogen to phosphorus ratio, MC: Mucilage cells, LA: Leaf area, LW: Leaf width, and LL: Leaf length.

variation was 19.34%. Among them, the variability in the diameter was the largest and the variability in the average clustering coefficient was the smallest.

### Relationship between leaf trait network of _P. euphratica_ and soil factors in the main stream of Tarim River

PCA analysis of the 11 soil factors (Fig 6) showed that the first four principal components accounted for 90.7% of the original information, whereas the first (PC1–1), second (PC1–2), third (PC1–3), and fourth (PC1–4) principal components explained 35.5, 29.1, 13.9, and 12.2% of the original information, respectively. SOM, WC, and STK were selected as

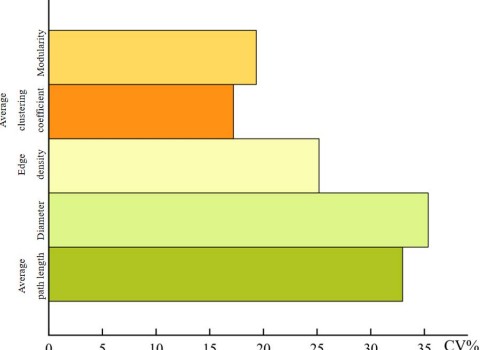

**Fig 5. The variability of network parameters in the main stream of Tarim River.**

representative indicators, and correlation analysis was performed using network parameters (Fig 7). The results showed that modularity was significantly negatively correlated with STK and SOM ($P < 0.05$).

**Relationship between leaf trait network of *P. euphratica* and climatic factors along the main stream of Tarim River**

The PCA analysis of the 19 climatic factors (Fig 8) showed that the first three principal components accounted for 87.21% of the original information, with PC1 (BIO3, BIO1, BIO12) explaining 60.89%, PC2 (BIO6, BIO11, BIO9) explaining 15.23%, and PC3 (BIO13, BIO16, BIO19) explaining 11.09%. These results indicated that PC1 and PC2 collectively

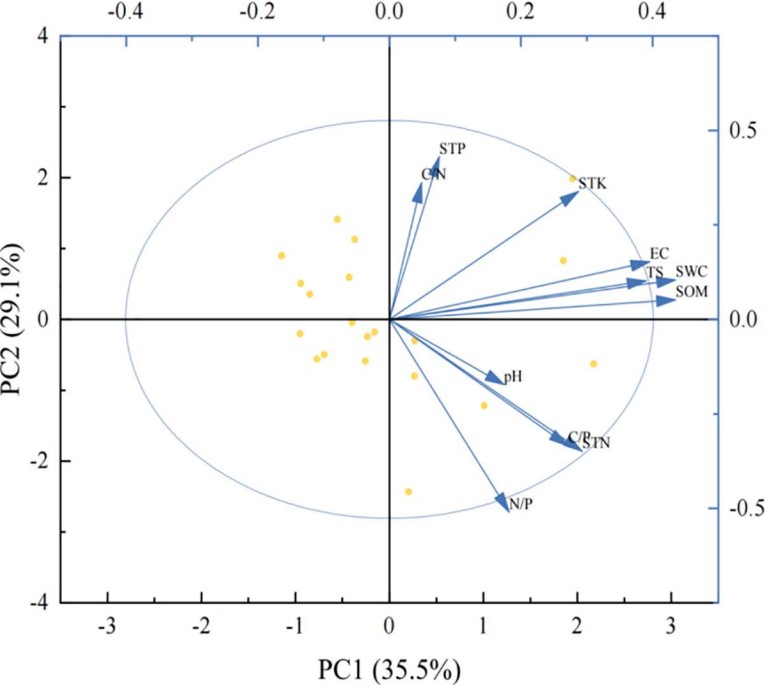

**Fig 6. PCA analysis of soil factors.** Note: C/N: Soil organic matter to nitrogen ratio, STP: Soil total phosphorus, STK: Soil total potassium, EC: Electrical conductivity, TS: Total salt, SWC: Water content, SOM: Soil total organic matter, pH: pH, C/P: Soil organic matter to phosphorus ratio, STN: Soil total nitrogen, and N/P: Soil nitrogen : phosphorus ratio.

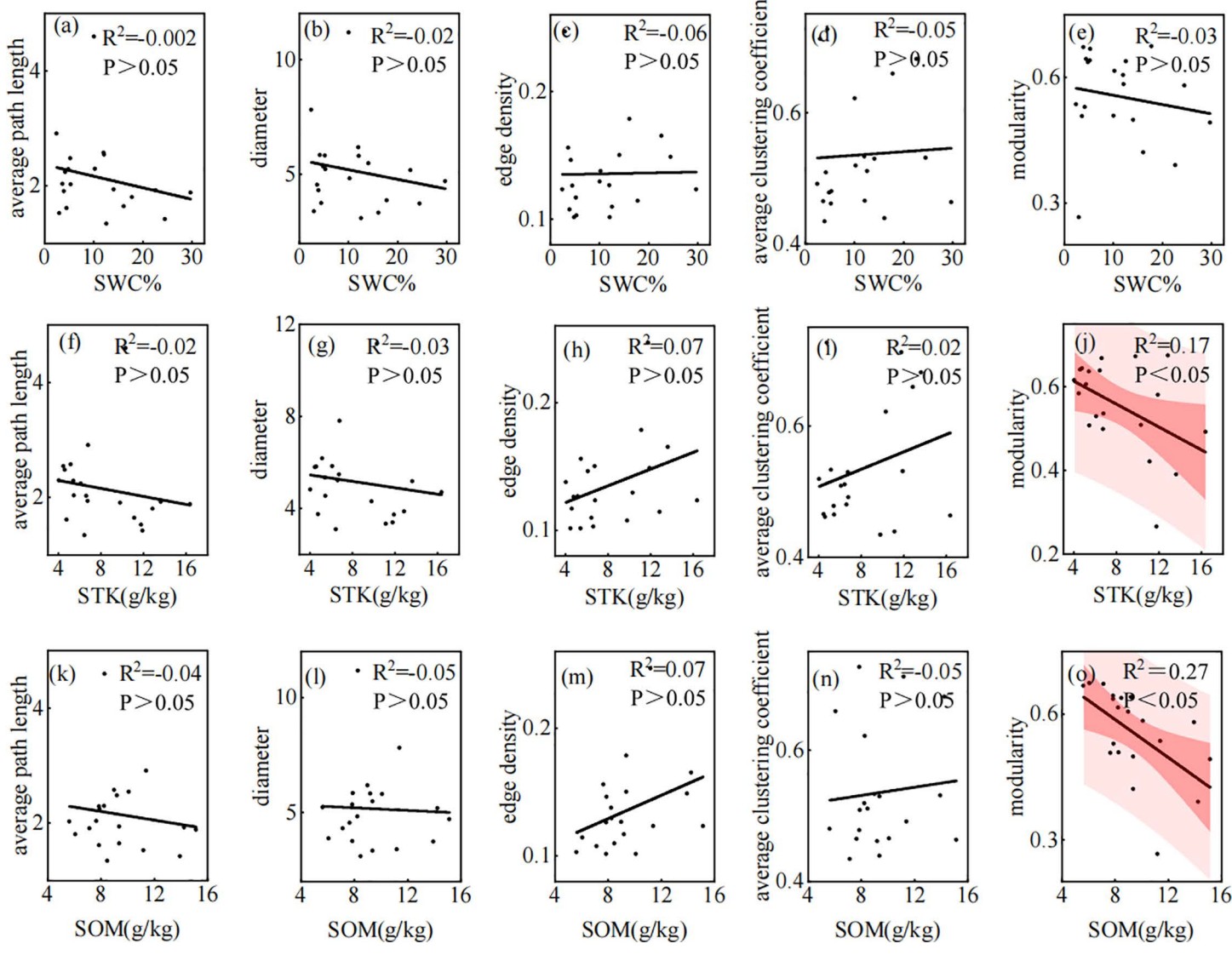

**Fig 7. The relationships between overall parameters of leaf traits network and soil factors.** Note: The shaded red area represents the 95% confidence interval and the black line represents the linear regression fit. SWC: Water content, STK: Soil total potassium, SOM: Soil total organic matter.

captured the majority of climatic variability across sampling sites. To further explore the collective influence of climatic variables on LTN parameters, we performed correlation analysis between PCs (BIO3, BIO6, and BIO13) and LTN characteristics (Fig 9). The results showed that the average clustering coefficient of the leaf trait network was significantly positively correlated with BIO03 ($P<0.05$); the average path length and diameter of the leaf trait network were significantly positively correlated with BIO06 ($P<0.05$), while the average clustering coefficient was significantly negatively correlated with BIO06 ($P<0.05$); additionally, the diameter of the leaf trait network was significantly positively correlated with BIO13 ($P<0.05$).

## Effects of climate and soil on network parameters of leaf traits

A systematic clustering analysis (using Euclidean distance and Ward's method) was performed on 20 sampling sites along the mainstream of the Tarim River in Xinjiang, China, based on 19 climatic factors and 11 soil factors. According to the

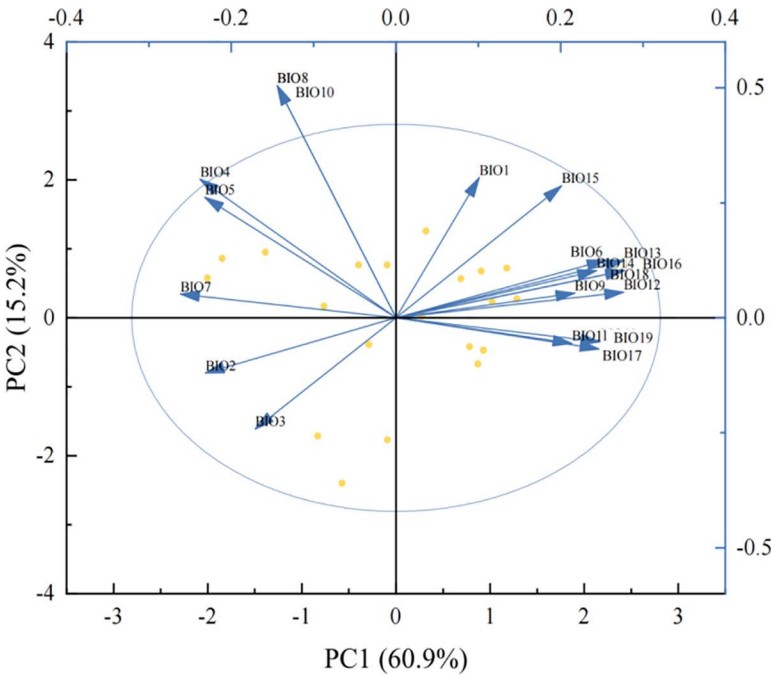

**Fig 8. PCA analysis of climate factors.** Note: BIO7: Temperature annual range, BIO5: Max temperature of warmest month, BIO4: Temperature seasonality, BIO8: Mean temperature of wettest quarter, BIO10: Mean temperature of warmest quarter, BIO1: Annual mean temperature, BIO15: Precipitation seasonality, BIO6: Min temperature of coldest month, BIO13: Precipitation of wettest month, BIO14: Precipitation of driest month, BIO16: Precipitation of wettest quarter, BIO18: Precipitation of warmest quarter, BIO9: Mean temperature of driest quarter, BIO12: Annual precipitation, BIO11: Mean temperature of coldest quarter, BIO19: Precipitation of coldest quarter, BIO17: Precipitation of driest quarter, BIO3: Isothermality, and BIO2: Mean diurnal range.

indicator characteristics and feature profiles of each cluster type, the sampling sites can be categorized into two major types: arid zones and extremely arid zones. This spatial pattern aligns highly with the environmental filtering theory, indicating that habitat heterogeneity drives the formation of modular community structures by screening adaptive traits (e.g., drought and salt tolerance). Hierarchical clustering further validated the hypothesis of synergistic environmental effects: the modularity index (Q = 0.78) of the LTN was significantly higher than that of single-trait analysis (Q = 0.52, $P < 0.01$)[26]. This demonstrates that multitrait synergistic responses (e.g., coexistence of drought and salt resistance traits) more accurately reflect the adaptive strategies of communities under environmental pressures (Fig 10).

The effects of climate and soil on the leaf trait network of *P. euphratica* were quantified using VPA (Fig 11). The results showed that climate, soil, and the interactions thereof had a common effect on the average path length, diameter, edge density, average clustering coefficient, and modularity. The combined effects of climate and soil explained 57% of the average path length. In the single effect, climate had a greater impact on the average path length, with an interpretation of 53.96%, whereas the single effect of soil was small, with an interpretation of 5.2%. In the interaction analysis, these two factors explained 2.1% of the variation. The combined effects of climate and soil accounted for 55% of the diameter. In the single effect, climate had a greater influence on diameter, explaining 50.53%. The single effect of soil was small, and the explanation was 4.35%. In the interaction analysis, these two factors explained 0.18% of the variation. The combined effects of climate and soil accounted for 71% of edge density. In the single effect, climate had a greater influence on edge density, with an explanatory degree of 50.73%. The single effect of soil was small, with an explanatory degree of 33.79%. In the interaction analysis, these two factors explained 13.34% of the variation. The combined effects of the climate and soil accounted for 70% of the average clustering coefficients. Among the individual effects, climate had the greatest

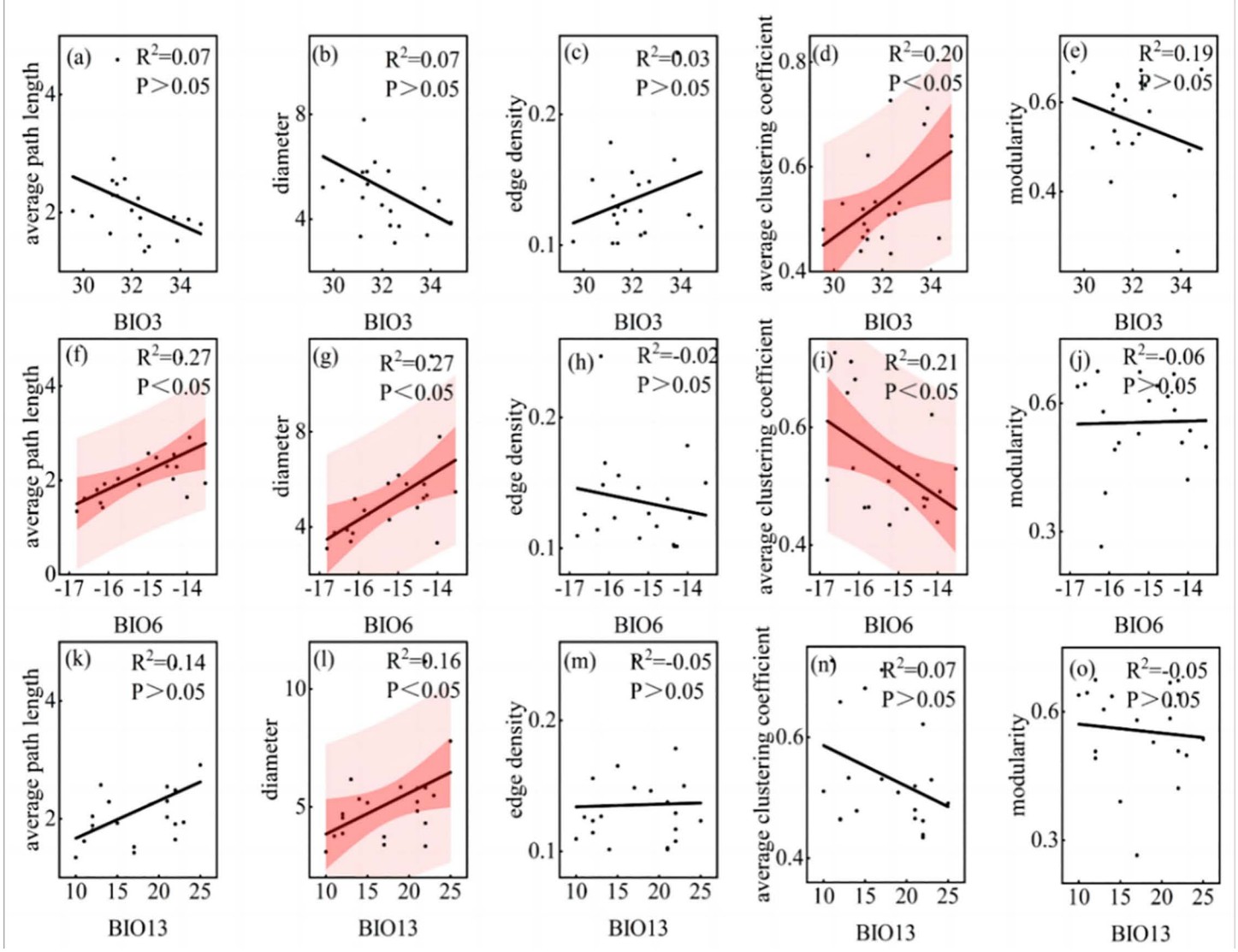

**Fig 9. Relationship between overall parameters of leaf traits network and climatic factors.** Note: The shaded red area represents the 95% confidence interval and the black line represents the linear regression fit. BIO3: Isothermality, BIO6: Min temperature of coldest month, BIO13: Precipitation of wettest month.

impact on the average clustering coefficient, with an interpretation degree of 63.47%. The individual effect of the soil was small, with an interpretation degree of 10.19%. In the interaction analysis, the two factors explained 3.36% of variation. The combined effects of the climate and soil accounted for 88% of the average clustering coefficients. Among the individual effects, soil had the greatest impact on the average clustering coefficient, with an interpretation degree of 50.45%. The individual effects of the climate were relatively small, with an interpretation degree of 43.90%. In the interaction analysis, these two factors explained 6.07% of the variation. Simultaneously, we found that the nodal parameters of some leaf traits in the LTN were also related to climate and soil factors (S6–S8 Tables in S1 File).

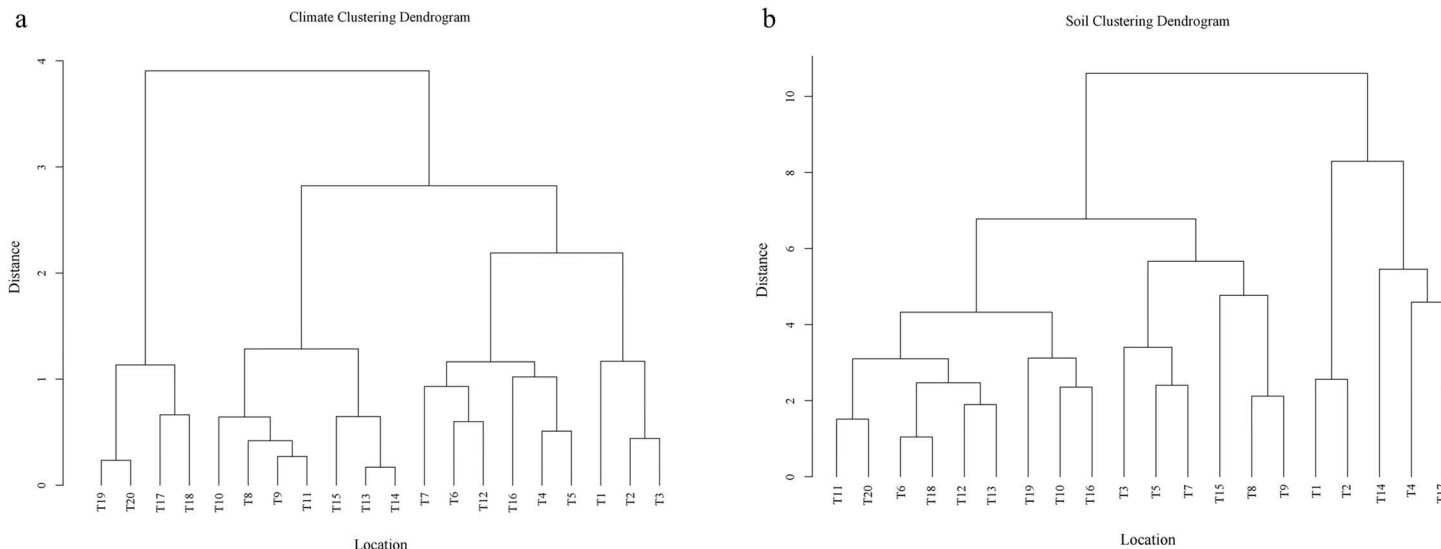

**Fig 10. Clustering of Climate-Soil Factors in the Tarim River Mainstream.**

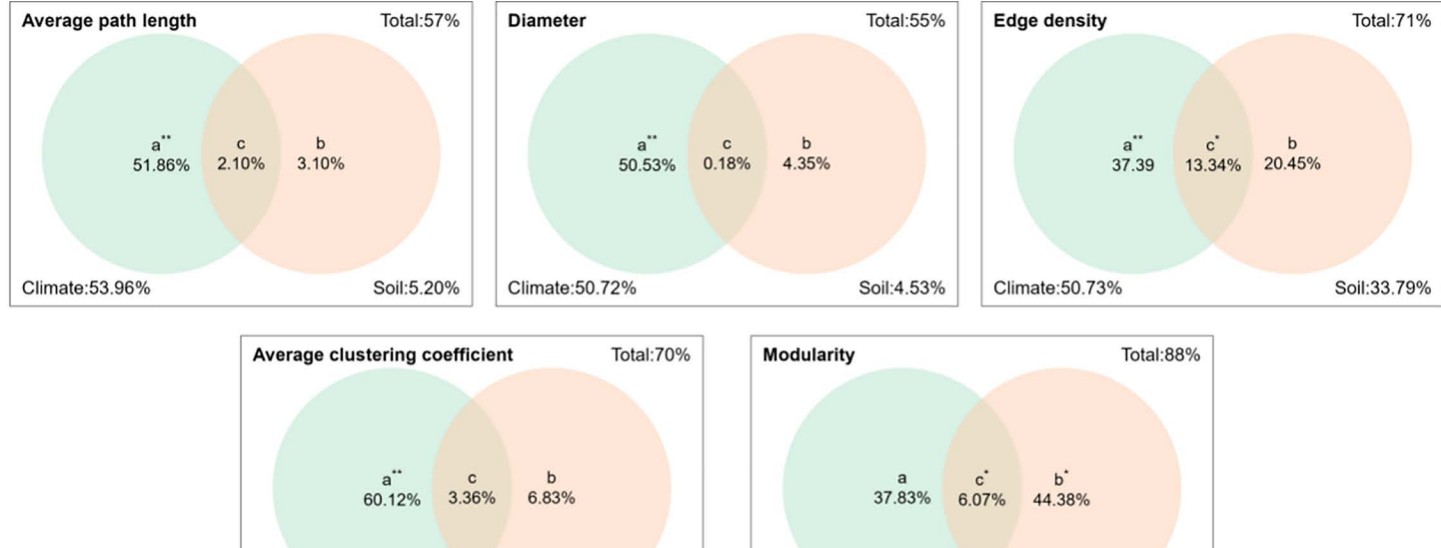

**Fig 11. Interpretation degree of climate and soil to network parameters.** Note: Divide the explanatory power of climate and soil on network parameters into individual effects and interaction effects, i.e., the independent effect of climate (a), the independent effect of soil (b), and the interaction effect between climate and soil (c).*$P$<0.05, **$P$<0.01.

## Discussion

### The relationship between leaf trait network of *P. euphratica* and soil factors in the main stream of Tarim River

In the main stream of the Tarim River, the variability in average path length and diameter was large, indicating that the interaction and connection between leaf traits of *P. euphratica* may be more diversified. The variability in edge density and modularity may be affected by the availability of resources and environmental conditions, such as climate and soil [37]. These factors may promote the leaf trait network to form a closer or more dispersed network structure to maximize the resource utilization efficiency. The variability of the average clustering coefficient indicates the local connectivity and trait tendency of leaf traits of *P. euphratica*, which reflects the heterogeneity of environmental conditions in the main stream of the Tarim River Basin and makes the leaf trait network of *P. euphratica* adopt different strategies when forming functional groups or modules.

WC, STK, and SOM are important indices that reflect soil fertility, fertilizer supply capacity, and plant growth status. The negative correlation between the average path length, diameter, modularity, WC, STK, and SOM indicated that when the content of WC, STK, and SOM was high, the connection degree of the leaf trait network increased, and the network structure was more concentrated. This indicates that in environments with abundant resources, the interaction between leaf traits is closer [27], and the network structure is more concentrated and efficient. Simultaneously, the decrease in modularity indicated that the functional modules were closely connected under better soil conditions, and the differences between different modules were reduced. Modularity was significantly negatively correlated with STK and SOM, indicating that these had a greater impact on the functional modules of the leaf trait network. Abundant soil resources reduced the modularity of the leaf trait network, reflecting the common utilization and interdependence of resources between the functional modules of leaf traits in *P. euphratica*. There was a positive correlation between WC, STK, SOM, edge density, average, and the clustering coefficient. This is likely because a resource-rich environment with high WC, STK, and SOM promotes direct interactions between leaf traits and builds a closer network structure.

### Relationship between leaf trait network of *P. euphratica* and climatic factors along the main stream of Tarim River

High isothermality indicates that the climate in the region is relatively stable [38]. Average path length, diameter, and modularity were negatively correlated with BIO3, indicating that the leaf trait network of *P. euphratica* tended to form a more compact and centralized structure in areas with relatively small temperature changes and relatively stable environments. This may be because under relatively stable temperature conditions, the interactions between traits are more frequent and the synergy is better, resulting in a decrease in the average path length and diameter, whereas higher isothermality may reduce niche differentiation and the modularity of the network [39]. The edge density and average clustering coefficient were positively correlated with BIO3, indicating that in an environment with small temperature fluctuations, the connectivity of the *P. euphratica* leaf trait network and the relationships between traits increased. There was a significant positive correlation between BIO3 and the average clustering coefficient, indicating that only specific traits, rather than all traits in the network, had better synergy and that the leaf traits of *P. euphratica* tended to form specific functional modules to achieve their functions. This may be an adaptation to resource utilization efficiency and survival strategies in the environment, reflecting the important relationship between the closely related and mutually supportive leaf traits of *P. euphratica*.

The average path length and diameter were significantly positively correlated with BIO6, which was likely due to the weakening of the synergy between the leaf network traits in the low-temperature environment, and the structure of the leaf trait network became more dispersed. *P. euphratica* adapts by adjusting its physiological and morphological characteristics. Edge density and average clustering coefficient were significantly negatively correlated with BIO6. This significant negative correlation indicates that *P. euphratica* may reduce the relationship between leaf traits in cold environments and aggregate traits into specific functional modules to resist low-temperature stress, thereby improving the ability to adapt to the environment.

Precipitation during the wettest months is the most significant climatic factor affecting tree height growth [40]. The positive correlation between BIO13 and average path length, diameter, and edge density showed that with an increase in the precipitation of the wettest month (BIO13), the overall independence of leaf traits increased. The average clustering coefficient and modularity were negatively correlated with BIO13, indicating that under high-precipitation conditions, the connections between traits were looser, the local aggregation of the leaf trait network was lower, and the network structure was more dispersed. With increasing rainfall, the leaf trait network module showed a trend of loose internal and tight external connections.

Climatic factors are the primary environmental drivers influencing network parameters such as average path length, diameter, edge density, and average clustering coefficient. Their explanatory power generally exceeds that of soil factors. The most significant impact was observed on the average clustering coefficient, explaining 63.47% of its variance. Although soil factors exhibit relatively smaller independent effects, they significantly influenced modularity ($P < 0.05$), indicating their non-negligible role in shaping certain leaf trait network parameters-particularly when interacting with climate, which can alter network structure. Overall, soil factors significantly influenced specific parameters; however, climatic factors remained dominant, confirming the climate-driven nature of inland river basin ecosystems in arid regions. Soil indirectly influenced functional differentiation by regulating modularity ($P < 0.05$), while the climate-soil interaction demonstrated a synergistic effect on edge density ($P < 0.05$). This synergy may arise from coupled water-nutrient mechanisms, such as osmotic regulation of soil total potassium under drought stress, which modulates trait associations. The strong correlations between principal components (PCs) and leaf trait network (LTN) parameters underscore that climatic effects on Populus euphratica are not isolated but operate through synergistic interactions among multiple variables.

## Conclusions

The average path length, diameter, and edge density in the leaf trait network of *P. euphratica* in the main stream of the Tarim River had a large degree of variation between 20 sampling points, and the overall leaf network parameters had a large spatial variability. Under the influence of different soil and climatic factors, the leaf trait network is promoted to form a closer or more dispersed network structure, thereby maximizing resource utilization efficiency.

Leaf trait networks were significantly affected by environmental factors. Average path length, diameter, and modularity were negatively correlated with water content, soil total potassium, and soil total organic matter, whereas edge density and average clustering coefficient were positively correlated with these soil factors. The negative correlations between modularity and soil total potassium and soil total organic matter were significant. In addition, these leaf trait network parameters were also related to climatic factors, among which average path length, diameter, and modularity were negatively correlated with isothermality and positively correlated with min temperature of coldest month. In contrast, the edge density and average clustering coefficient were positively correlated with isothermality but negatively correlated with min temperature of coldest month, and the correlation between the average clustering coefficient and these two climatic factors was significant. The average path length, diameter, and edge density of the leaf trait network were positively correlated with precipitation of wettest month, whereas the average clustering coefficient and modularity were negatively correlated with precipitation of wettest month, and the diameter was positively correlated with precipitation of wettest month.

Climatic factors play a decisive role in shaping the network structure of leaf traits and are the main environmental factors affecting the network parameters of leaf traits of *P. euphratica*. The degree of interpretation of individual effects was generally higher than that of the soil factors; However, soil factors had the greatest influence on the modularity when acting alone. The interactions between soil and climatic factors contributed to all leaf trait network parameters. Although the interpretation was relatively small, climatic and soil factors alone did not affect the leaf trait network. Interactions between these factors also shaped the spatial variability of leaf traits to a certain extent. However, compared to soil factors, climatic factors can significantly change the structure of the leaf trait network. Crucially, the variation in the *P. euphratica* leaf trait network was mainly driven by climate factors.

## Supporting information

**S1 File. All the supporting tables in this article.**

(ZIP)

## Author contributions

**Data curation:** Chengzhi Peng, Shiyu Yao, Jie Wang.

**Formal analysis:** Chengzhi Peng, Shiyu Yao.

**Funding acquisition:** Wenjuan Huang.

**Investigation:** Chengzhi Peng, Shiyu Yao, Jie Wang, Shuangfei Song, Pei Zhang.

**Methodology:** Chengzhi Peng, Shiyu Yao, Wenjuan Huang.

**Resources:** Wenjuan Huang, Peipei Jiao.

**Software:** Chengzhi Peng, Shiyu Yao, Jie Wang, Shuangfei Song, Pei Zhang.

**Writing – original draft:** Chengzhi Peng, Shiyu Yao.

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
