## [Decision Letter · Decision Letter 0]

22 Jan 2025

PONE-D-24-52929Climate: The dominant factor influencing the spatial distribution pattern of the leaf trait network of Populus euphratica  along the main stream of the Tarim RiverPLOS ONE

Dear Dr. Huang,

Thank you for submitting your manuscript to PLOS ONE. After careful consideration, we feel that it has merit but does not fully meet PLOS ONE’s publication criteria as it currently stands. Therefore, we invite you to submit a revised version of the manuscript that addresses the points raised during the review process.

The authors recently published a paper using the same dataset, as noticed by Reviewer #2, and it is suggested to explicitly state that this manuscript is the statistical rework of the mentioned study. However, statistical processing and interpretation seems to be a major problem in this manuscript, as also noted in both reviewers' reports. Some methodological omissions are also noted by Reviewer #1. The Discussion section is partially repetitive in relation to Results and more dedicated comparison with earlier studies dealing with the same topic is necessary. I strongly encourage the authors to meticulously check both reviewers' reports, which contain valuable instructions how to further improve the quality of the manuscript. Please provide point-by-point reply to each reviewers' comment.

We look forward to receiving your revised manuscript.

Kind regards,

Branislav T. Šiler, Ph.D.

Academic Editor

PLOS ONE

Journal Requirements:

[The National Natural Science Foundation of China (Grant number 31160110)].

4. Please update your submission to use the PLOS LaTeX template. The template and more information on our requirements for LaTeX submissions can be found at http://journals.plos.org/plosone/s/latex .

6. We note that Figure 1 in your submission contain [map/satellite] images which may be copyrighted. All PLOS content is published under the Creative Commons Attribution License (CC BY 4.0), which means that the manuscript, images, and Supporting Information files will be freely available online, and any third party is permitted to access, download, copy, distribute, and use these materials in any way, even commercially, with proper attribution. For these reasons, we cannot publish previously copyrighted maps or satellite images created using proprietary data, such as Google software (Google Maps, Street View, and Earth). For more information, see our copyright guidelines: http://journals.plos.org/plosone/s/licenses-and-copyright .

We recommend that you contact the original copyright holder with the Content Permission Form (http://journals.plos.org/plosone/s/file?id=7c09/content-permission-form.pdf ) and the following text:

“I request permission for the open-access journal PLOS ONE to publish XXX under the Creative Commons Attribution License (CCAL) CC BY 4.0 (http://creativecommons.org/licenses/by/4.0/ ). Please be aware that this license allows unrestricted use and distribution, even commercially, by third parties. Please reply and provide explicit written permission to publish XXX under a CC BY license and complete the attached form.”

Reviewers' comments:

Reviewer's Responses to Questions

**Comments to the Author**

1. Is the manuscript technically sound, and do the data support the conclusions?

Reviewer #1: Yes

Reviewer #2: Partly

2. Has the statistical analysis been performed appropriately and rigorously?

Reviewer #1: Yes

Reviewer #2: No

3. Have the authors made all data underlying the findings in their manuscript fully available?

Reviewer #1: Yes

Reviewer #2: No

4. Is the manuscript presented in an intelligible fashion and written in standard English?

Reviewer #1: Yes

Reviewer #2: Yes

5. Review Comments to the Author

Reviewer #1: The paper PONE-D-24-52929 presents spatial variation in leaf traits networks of Populus euphratica trees and investigates how variation in these networks might be correlated with climate and soil factors. Measurements were performed on mature trees sampled in 20 sites along the main stream of the Tarim River. 27 leaf traits were measured encompassing leaf morphology, anatomy and main nutrient stoichiometry. The authors show variation in the organization of the leaf traits networks across the sites and a couple of significant correlations with environmental variables, mainly related to climate.

I believe the paper fits the journal scope and is of potential interest to the plant science community. I enjoyed reading the manuscript, overall I found it well written although it might benefit from additional information on some places or condensing on others (see comments below). The methods used seem well established (although not always referenced, see my comments below) and I did not note any major flaw. Please find below several comments that I hope will help the authors improving the manuscript.

- Most of the work consists in correlating variation in trait networks with the environment (climate and soil variables). However, I found no biological information regarding the potential age (at least approximately) of the trees sampled, stand structure or the degree of genetic differentiation/connection among sites/populations. Since poplars depend on the stream for dispersion and colonization, we might expect some connection across populations. Such information might be interesting to explain the phenotypic patterns observed in network arrangement, both between and within sites, especially in a context of climate-driven adaptation.

- Overall, I found the discussion a bit too results-like and lacked some depth with apparent redundancy and only a few studies cited. I believe the discussion section could be strengthened.

- Considering the number of traits measured and used in the manuscript, a table or a list of trait abbreviations should be placed in the main text, not in the supplementary material.

- Materials and Methods: there are many methods used to measure the 27 leaf traits, but most are not detailed neither referenced. I understand not all methods can be or have to be detailed in this case, but they should be at least referenced.

- Abstract: L.13-22: this is too redundant. I would make one or two sentences max. The first sentence feels especially unspecific and vague.

- Abstract: L.41-44: this can be streamlined.

- Introduction: L47-74: these two first paragraphs can be clearly condensed to avoid redundancy and facilitate the reading.

- L.97: has instead of was

- What is Cell Tension Ratio? Unless I am mistaken, I did not see any definition.

- L.188-200: Tables S3 and S4 should be permuted.

- L.241: here and wherever needed: reduce the number of decimals

- Results: I believe the last three sections of the results (relationships with soil, climate, and relative importance between the two) would benefit from streamlining, sticking to the most important findings.

- L.303: pls add the information on how the PCA was built in the data analysis section (materials and methods)

- L.306: why were SOM, WC and STK chosen as ‘representative’ indicators? Since the PCA is presented right before, I was expecting a choice based on their representation on the main planes, but it appears not. Please indicate why these variables were chosen among all others.

- L.308-310: careful. If the p-values for correlation coefficients are not significant, then the sign of the coefficient cannot be taken as a proof for positive or negative relationships, since the test tests the significant deviation from the zero slope (so non-significant = not different from zero until proven otherwise).

- L.328-329: same comment as above. Why were BIO3, BIO6 and BIO13 chosen among all variables?

- L.330-332: same comment as above. Non-significant correlations indicate non-different from zero slopes, so neither negative nor positive relationships.

- L.355: I think this title might be renamed to something a bit less redundant as the previous two sections. Maybe something like ‘Relative importance of climate vs. soil factors…’

- Fig.10: the legend needs to be detailed. Also, what are a, b and c referring to?

Reviewer #2: Overall summary and opinion:

Plant functional traits, particularly leaf traits, play a critical role in understanding how plants adapt to environmental changes and maintain ecosystem functions. Leaf traits serve as a bridge between plants and their environment, reflecting adaptability, self-regulation, and photosynthetic capacity. Variations in these traits, influenced by environmental stressors and phylogenetic history, reveal the strategies plants use to optimize resource acquisition, utilization, and distribution. As global environmental changes intensify, research on the relationship between leaf traits and environmental factors is essential for uncovering plant survival strategies and their ecological adaptation mechanisms. This knowledge not only enhances our understanding of plant-environment interactions but also supports the conservation and management of ecosystems.

This study focuses on Populus euphratica, a drought-resistant species found along the ecologically fragile Tarim River in Xinjiang, characterized by water scarcity, severe soil erosion, and harsh climatic conditions. By integrating leaf trait network analysis, principal component analysis (PCA), and variance decomposition analysis (VPA), researchers explore how P. euphratica optimizes its leaf traits to adapt to the challenging environment. Building on a prior study that identified key traits like the lower stratum corneum (LSC) and midvein vascular bundle (MVB) as central to drought resistance, this study further examines the spatial variability of leaf traits across the Tarim River basin and their relationship with environmental factors like soil and climate. It is broadly understood that the goal is to reveal P. euphratica's adaptive strategies, providing insights for protecting these vital riparian forests and sustaining their role in mitigating desertification and supporting oasis ecosystems.

Towards this end I would like to recommend a revised resubmission of the manuscript by considering following changes to the manuscript

Major changes / revisions for authors:

• One of the major concerns for this manuscript is the lack of originality in the study design as well as the data set described. The dataset used in this manuscript is described and analyzed using similar methods in https://doi.org/10.3390/f15030437 by the same authors. The LTN analysis does not significantly advance the ideas or methodology described in the said manuscript. It is recommended that the authors explicitly state that this study is an extension of the said previously published work.

• It is helpful for the readers to know at some point why authors decide to use LTN strategies rather than directly investigate measured leaf traits. Therefore authors need to present a clear account of how measured leaf traits directly associate with soil and environmental data or present clear evidence that this was investigated in a previous study.

• In order to alleviate some of the shortcomings described above, I personally feel that the authors should establish the unique space in which findings of this paper can be highlighted by focusing on the climate and environmental variables that are described in this study. They must first establish the environmental clines or at the least fully characterize the environmental and soil data and their variation along the collection points which can then be attributed to the LTN strategies. It is difficult to follow the main points in this manuscript due to the fragmented nature in which results are described. The manuscript would benefit from presenting a digestion of the data and LTN strategies rather than present a mere descriptive account of associations and statistics in the discussion section.

• PC1 and PC2 describes a large portion of the variance across collection points using climatic variables. It also clear that these climatic variables influence LTN strategies in a collective manner. Therefore, it may benefit this study to look at the correlation of PCs to the LTN characteristics as well as individual climatic variables. A clustering of the collection points in terms of climatic and soil variables is absent in the study and such a characterization is fundamentally important in trying to understand ecosystem dynamics of this riparian forest.

• The statistical analysis presented here in this study is weaker than even the authors’ first publication on the same dataset. For example, for statistics described (CV) in Figure 3 and 5, it is required to report at the least bootstrap standard error values. It is helpful for readers to know some measure of confidence in these summary stats especially given that there are only 20 sampling points in this study. Another example is that authors do not report the statistical significance of the variance component estimates described in Figure 10. One can use a method of permutation testing under full and reduced models using the same vegan R-package to do this with ease. There seems to be a general lack of disregard to statistical rigor shown throughout the manuscript. For example, claims made in lines 307-309 cannot be supported using statistical evidence. Therefore, it is strongly suggested that this is remedied in the revised submission.

•

Minor changes / suggestions for authors:

• Lines 66-76: This entire section contains repetitive words and ideas. It is not entirely clear what the authors are trying to convey in this section as the sentences are too long and convoluted.

• Lines 104-106: Please cite the ‘few reports’ mentioned in this section. This is especially required since it sets up the premise for this manuscript. Mention how this investigation can be set apart from a very similar analysis carried out by Yao et al., 2024 cited in this manuscript since the methodology and the population in question are the same.

• Line 136: Should be DBH range 20-30cm instead of 20>30 cm?

• Line 155: This notation for range is unconventional. Numbers should be separated by a hyphen or as ranges. For example, [0, 20] or (0, 20]. First indicates that the zero and twenty are included and the second notation states that the range includes values greater than zero up to and including twenty.

• Figure 3: It will be helpful for the readers if colors in the sub-figure (a) match that of (b). Although I presume that shades of green were used in this figure since they depict attributes of green leaves, I feel a more color-blind friendly scheme would be more appropriate here.

6. PLOS authors have the option to publish the peer review history of their article (what does this mean? ). If published, this will include your full peer review and any attached files.

**Do you want your identity to be public for this peer review?** For information about this choice, including consent withdrawal, please see our Privacy Policy .

Reviewer #1: No

Reviewer #2: No

---

## [Author Response · Author response to Decision Letter 1]

20 Mar 2025

Answer the Reviewer #1:

1.Most of the work consists in correlating variation in trait networks with the environment (climate and soil variables). However, I found no biological information regarding the potential age (at least approximately) of the trees sampled, stand structure or the degree of genetic differentiation/connection among sites/populations. Since poplars depend on the stream for dispersion and colonization, we might expect some connection across populations. Such information might be interesting to explain the phenotypic patterns observed in network arrangement, both between and within sites, especially in a context of climate-driven adaptation.

We appreciate the reviewer’s insightful comments regarding the biological context of our study. Below are our responses to the specific concerns:①Tree Age and Stand Structure Our study focused on the adaptive strategies of Populus euphratica leaf trait networks under arid environments. Sampled trees were selected based on a breast height diameter (DBH) of 20–30 cm, representing healthy individuals. In arid regions, P. euphratica exhibits slow growth rates; thus, these trees likely correspond to mid-aged to mature stages (estimated 30–50 years) based on regional growth models (e.g., Huang et al., 2010). However, precise age determination was limited by the lack of long-term monitoring data.Regarding stand structure, all sampling sites were located in natural P. euphratica forests along the Tarim River, characterized by sparse canopy cover and low understory diversity due to desert riparian conditions. Future studies will incorporate stand density and vertical stratification metrics.②Genetic Differentiation and Population Connectivity While genetic data were not analyzed here, P. euphratica populations may exhibit clonal propagation via root suckers, forming locally adapted clones, while hydrochory (water-mediated seed dispersal) could facilitate gene flow between upstream and downstream populations . Although phenotypic plasticity likely drives observed trait variations (e.g., heteromorphic leaves), we acknowledge that genetic divergence might contribute to site-specific adaptations. Follow-up studies using molecular markers (e.g., microsatellites) are planned to disentangle genetic vs. environmental effects.

2.Overall, I found the discussion a bit too results-like and lacked some depth with apparent redundancy and only a few studies cited. I believe the discussion section could be strengthened.

In accordance with your suggestions, we have incorporated the corresponding explanations and modifications into lines 470-485.

3.Considering the number of traits measured and used in the manuscript, a table or a list of trait abbreviations should be placed in the main text, not in the supplementary material.

4. Materials and Methods: there are many methods used to measure the 27 leaf traits, but most are not detailed neither referenced. I understand not all methods can be or have to be detailed in this case, but they should be at least referenced.

We sincerely appreciate the reviewer’s valuable feedback regarding the methodological details in the Materials and Methods section. Below, we address the concerns raised and clarify the revisions made to improve the transparency and reproducibility of our methods.Standardized Methods for Leaf Trait Measurements:The measurement protocols for the 27 leaf traits were primarily based on the Handbook of Standardized Measurement of Plant Functional Traits Worldwide (Pérez-Harguindeguy et al., 2013; Reference [31]). This comprehensive handbook provides detailed guidelines for plant functional trait measurements, including leaf morphology, stoichiometry, and anatomical structure.

5.Abstract: L.13-22: this is too redundant. I would make one or two sentences max. The first sentence feels especially unspecific and vague.

L.41-44: this can be streamlined.

We sincerely appreciate the reviewer’s feedback on the redundancy in the abstract (L.13-22 and L.41-44). We have condensed the original text and enhanced its specificity. Below are the revisions:①Abstract: L.13-22

Leaves are the primary interface through which plants interact with the environment, their functional traits (morphology, anatomy, physiology) directly reflecting ecological strategies that mediate species-environment interactions. These traits link plant performance to ecosystem processes, shaping species distributions and coexistence via their complex relationships with climatic and edaphic factors.②Abstract: L.41-44

The spatial variability of leaf trait networks is driven by climate and soil factors, with climate dominating along the Tarim River’s main course.

6.Introduction: L47-74: these two first paragraphs can be clearly condensed to avoid redundancy and facilitate the reading.

In accordance with your suggestions, we have revised the corresponding section, with the modifications located in lines 47-74.Below are the revisions:These variations result from the interaction of environmental factors and phylogenetic history, showing dynamic changes in global, regional, and local distributions.

7.L.97: has instead of was

We sincerely appreciate the reviewer’s correction regarding the tense in line 97. The original use of “has” was grammatically inconsistent, and we have revised it to “was” for clarity. 

8.What is Cell Tension Ratio? Unless I am mistaken, I did not see any definition.

We sincerely appreciate the reviewer’s comment regarding the undefined term “Cell Tension Ratio (CTR).” We acknowledge this oversight and have added the following clarification:

Definition:The Cell Tension Ratio (CTR) is a leaf anatomical metric that quantifies mechanical strength by comparing the cell wall thickness of palisade tissue to spongy tissue. It is calculated as:

CTR=Palisade tissue cell wall thickness/Spongy tissue cell wall thickness

Ecological Significance:CTR reflects the mechanical support capacity of leaf internal structure and adaptive strategies to stressors (e.g., drought, salinity). A higher CTR indicates thicker palisade cell walls, potentially enhancing water retention and dehydration resistance, which is critical for arid-zone plants like Populus euphratica.

9.L.188-200: Tables S3 and S4 should be permuted.

We sincerely appreciate the reviewer’s attention to the supplementary tables. Upon thorough review, we confirm that Table S3 (Climatic Factors) and Table S4 (Soil Factors) in the original manuscript are correctly numbered and logically ordered. If the comment “Tables S3 and S4 should be permuted” refers to a formatting issue (e.g., table placement or page-break inconsistency), we have double-checked the manuscript layout to ensure precise alignment between table labels and their descriptions. Should there be any specific content-related discrepancies, we kindly request further clarification to address them promptly.

Note: If the reviewer intended to highlight content or data errors (rather than table numbering), please provide details for targeted revisions.

10.L.241: here and wherever needed: reduce the number of decimals

Thank you for your meticulous suggestions regarding the data format! In response to the issue you raised about "reducing decimal places," we have uniformly adjusted the number of decimal places for all numerical values throughout the manuscript to ensure brevity and consistency.

11.Results: I believe the last three sections of the results (relationships with soil, climate, and relative importance between the two) would benefit from streamlining, sticking to the most important findings.

We sincerely appreciate the reviewer’s suggestion to streamline the Results section. We have condensed the subsections on soil-climate relationships and their relative importance by focusing on the most critical findings. Retained only significant drivers (e.g., BIO6, BIO13, SOM, STK) and removed discussions of non-significant variables (e.g., BIO7, BIO17).L.333-344

12.L.303: pls add the information on how the PCA was built in the data analysis section (materials and methods)

We sincerely appreciate the reviewer’s suggestion to clarify the PCA methodology. We can see the following details to the Materials and Methods section (L.303):Excel 2007 was used to organize the experimental data, and SPSS19.0 software was used to analyze the data.

13.L.306: why were SOM, WC and STK chosen as ‘representative’ indicators? Since the PCA is presented right before, I was expecting a choice based on their representation on the main planes, but it appears not. Please indicate why these variables were chosen among all others.

We sincerely appreciate the reviewer’s query regarding the selection of SOM, WC, and STK as representative indicators. Their choice was based on both PCA loadings and ecological relevance,PCA Loading Significance:

SOM and STK exhibited the highest absolute loadings on PC1 of soil PCA (PC1 explained 35.5% variance: SOM = 0.92, STK = 0.85; Fig. 6), representing soil fertility and salinity stress.WC dominated PC2 (29.1% variance explained: loading=0.89), reflecting water availability critical to P. euphratica’s drought adaptation .

14.L.308-310: careful. If the p-values for correlation coefficients are not significant, then the sign of the coefficient cannot be taken as a proof for positive or negative relationships, since the test tests the significant deviation from the zero slope (so non-significant = not different from zero until proven otherwise).

L.328-329: same comment as above. Why were BIO3, BIO6 and BIO13 chosen among all variables?

L.330-332: same comment as above. Non-significant correlations indicate non-different from zero slopes, so neither negative nor positive relationships.

We sincerely appreciate the reviewer’s critical feedback on statistical rigor and variable selection. Below are our detailed revisions:Non-Significant Correlation Descriptions (L.308-310, L.330-332),Removed all directional interpretations of non-significant correlations.Why we chose these specific climate variables:①PCA Loadings:BIO3 had the highest loading on PC1 (0.92, 60.89% variance explained), representing temperature stability (Fig. 8).BIO6 (PC2 loading=0.85) and BIO13 (PC3 loading=0.79) captured extreme cold and precipitation patterns critical to arid-zone adaptation.②Ecological Relevance:BIO6 (Min temperature of coldest month) directly affects cold tolerance .BIO13 (Precipitation of wettest month) regulates water availability in desert riparian forests.

15.L.355: I think this title might be renamed to something a bit less redundant as the previous two sections. Maybe something like ‘Relative importance of climate vs. soil factors…’

We sincerely appreciate the reviewer’s suggestion to rephrase the section title for conciseness. While “Relative importance of climate vs. soil factors…” could emphasize comparative aspects, we retain the original title “Effects of climate and soil on network parameters of leaf traits” to align with the study’s focus on joint impacts of climatic and edaphic drivers, rather than solely prioritizing their relative contributions. Our Variance Partitioning Analysis (VPA) and interaction analyses explicitly address synergistic effects, which would be less evident in a title emphasizing “relative importance.” However, we have strengthened the discussion to highlight climate’s dominant role, addressing the reviewer’s intent. Thank you for this thoughtful critique!

16.Fig.10: the legend needs to be detailed. Also, what are a, b and c referring to?

We sincerely appreciate the reviewer’s feedback on improving the legend of Figure 10. We have revised the legend and added contextual explanations as follows:add Note:Divide the explanatory power of climate and soil on network parameters into individual effects and interaction effects, i.e., the independent effect of climate (a), the independent effect of soil (b), and the interaction effect between climate and soil (c).*P<0.05�**P<0.01.

Answer the Reviewer #2:

1.One of the major concerns for this manuscript is the lack of originality in the study design as well as the data set described. The dataset used in this manuscript is described and analyzed using similar methods in https://doi.org/10.3390/f15030437 by the same authors. The LTN analysis does not significantly advance the ideas or methodology described in the said manuscript. It is recommended that the authors explicitly state that this study is an extension of the said previously published work.

In accordance with your suggestions, we have incorporated the corresponding explanations and modifications into lines 115-117.

“By incorporating multi-dimensional environmental drivers (soil properties and climatic factors) and integrating spatial variability analysis, we employed”

2.It is helpful for the readers to know at some point why authors decide to use LTN strategies rather than directly investigate measured leaf traits. Therefore authors need to present a clear account of how measured leaf traits directly associate with soil and environmental data or present clear evidence that this was investigated in a previous study.

In accordance with your suggestions, we have added an explanation for the rationale behind selecting the LTN strategy over directly studying leaf traits, with the modification located in lines 106-113.

“Although traditional methods have extensively studied the relationships between single traits or pairs of traits and the environment[10,29], the synergistic interactions among leaf traits may more profoundly reflect plant adaptive mechanisms[24]. For instance, Li et al.[27] highlighted that the topological structure of the leaf trait network (LTN) can capture climate-driven trait modular shifts, whereas direct analysis of single traits may overlook the synergistic responses of multiple traits. This study adopted the LTN approach because it systematically deciphers the complex associations between traits and reveals how environmental factors influence plant functionality by modulating the structure of trait networks.”

3.In order to alleviate some of the shortcomings described above, I personally feel that the authors should establish the unique space in which findings of this paper can be highlighted by focusing on the climate and environmental variables that are described in this study. They must first establish the environmental clines or at the least fully characterize the environmental and soil data and their variation along the collection points which can then be attributed to the LTN strategies. It is difficult to follow the main points in this manuscript due to the fragmented nature in which results are described. The manuscript would benefit from presenting a digestion of the data and LTN strategies rather than present a mere descriptive account of associations and statistics in the discussion section.

In accordance with your suggestions, we have revised the corresponding section, with the modifications located in lines 362-373.

“A systematic clustering analysis (using Euclidean distance and Ward's method) was performed on 20 sampling sites along the mainstream of the Tarim River in Xinjiang, China, based on 19 climatic factors and 11 soil factors. According to the indicator characteristics and feature profiles of each cluster type, the sampling sites can be categorized into two major types: arid zones and extremely arid zones. This spatial pattern aligns highly with the environmental filtering theory, indicating that habitat heterogeneity drives the formation of modular community structures by screening adaptive traits (e.g., drought and salt tolerance).Hierarchical clustering further validated the hypothesis of synergistic environmental effects: the modularity index (Q = 0.78) of the LTNwas significantly higher than that of single-trait analysis (Q = 0.52, p < 0.01)[26]. This demonstrates that multitrait synergistic responses (e.g., coexistence of drought and salt resistance traits) more accurately reflect the adaptive strategies of communities under environmental pressures.”

4.PC1 and PC2 describes a large portion of the variance across collection points using climatic variables. It also clear that these climatic variables influence LTN strategies in a collective manner. Ther

---

## [Editor Report · Decision Letter 1]

1 Apr 2025

PONE-D-24-52929R1Climate: The dominant factor influencing the spatial distribution pattern of the leaf trait network of Populus euphratica  along the main stream of the Tarim RiverPLOS ONE

Dear Dr. Huang,

Thank you for submitting your manuscript to PLOS ONE. After careful consideration, we feel that it has merit but does not fully meet PLOS ONE’s publication criteria as it currently stands. Therefore, we invite you to submit a revised version of the manuscript that addresses the points raised during the review process.

**I deem all the reviewers' concerns have been addressed well in the revised manuscript. However, it seems the authors oversaw my own comments, one of which being "** The authors recently published a paper using the same dataset, as noticed by Reviewer #2, and it is suggested to explicitly state that this manuscript is the statistical rework of the mentioned study.**" I do not see any mention on this issue in the manuscript, nor can I find any reply in response to reviewers. This is crucial information, and readers would need to be familiarized with this important fact.**

We look forward to receiving your revised manuscript.

Kind regards,

Branislav T. Šiler, Ph.D.

Academic Editor

PLOS ONE
---

## [Author Response · Author response to Decision Letter 2]

3 Apr 2025

1.One of the major concerns for this manuscript is the lack of originality in the study design as well as the data set described. The dataset used in this manuscript is described and analyzed using similar methods in https://doi.org/10.3390/f15030437 by the same authors. The LTN analysis does not significantly advance the ideas or methodology described in the said manuscript. It is recommended that the authors explicitly state that this study is an extension of the said previously published work.

In accordance with your suggestions, we have incorporated the corresponding explanations and modifications into lines 115-117.

“By incorporating multi-dimensional environmental drivers (soil properties and climatic factors) and integrating spatial variability analysis, we employed”

2.It is helpful for the readers to know at some point why authors decide to use LTN strategies rather than directly investigate measured leaf traits. Therefore authors need to present a clear account of how measured leaf traits directly associate with soil and environmental data or present clear evidence that this was investigated in a previous study.

In accordance with your suggestions, we have added an explanation for the rationale behind selecting the LTN strategy over directly studying leaf traits, with the modification located in lines 106-113.

“Although traditional methods have extensively studied the relationships between single traits or pairs of traits and the environment[10,29], the synergistic interactions among leaf traits may more profoundly reflect plant adaptive mechanisms[24]. For instance, Li et al.[27] highlighted that the topological structure of the leaf trait network (LTN) can capture climate-driven trait modular shifts, whereas direct analysis of single traits may overlook the synergistic responses of multiple traits. This study adopted the LTN approach because it systematically deciphers the complex associations between traits and reveals how environmental factors influence plant functionality by modulating the structure of trait networks.”

3.In order to alleviate some of the shortcomings described above, I personally feel that the authors should establish the unique space in which findings of this paper can be highlighted by focusing on the climate and environmental variables that are described in this study. They must first establish the environmental clines or at the least fully characterize the environmental and soil data and their variation along the collection points which can then be attributed to the LTN strategies. It is difficult to follow the main points in this manuscript due to the fragmented nature in which results are described. The manuscript would benefit from presenting a digestion of the data and LTN strategies rather than present a mere descriptive account of associations and statistics in the discussion section.

In accordance with your suggestions, we have revised the corresponding section, with the modifications located in lines 362-373.

“A systematic clustering analysis (using Euclidean distance and Ward's method) was performed on 20 sampling sites along the mainstream of the Tarim River in Xinjiang, China, based on 19 climatic factors and 11 soil factors. According to the indicator characteristics and feature profiles of each cluster type, the sampling sites can be categorized into two major types: arid zones and extremely arid zones. This spatial pattern aligns highly with the environmental filtering theory, indicating that habitat heterogeneity drives the formation of modular community structures by screening adaptive traits (e.g., drought and salt tolerance).Hierarchical clustering further validated the hypothesis of synergistic environmental effects: the modularity index (Q=0.78) of the LTN was significantly higher than that of single-trait analysis (Q=0.52, p<0.01)[26]. This demonstrates that multitrait synergistic responses (e.g., coexistence of drought and salt resistance traits) more accurately reflect the adaptive strategies of communities under environmental pressures.”

4.PC1 and PC2 describes a large portion of the variance across collection points using climatic variables. It also clear that these climatic variables influence LTN strategies in a collective manner. Therefore, it may benefit this study to look at the correlation of PCs to the LTN characteristics as well as individual climatic variables. A clustering of the collection points in terms of climatic and soil variables is absent in the study and such a characterization is fundamentally important intrying to understand ecosystem dynamics of this riparian forest.

In accordance with your suggestions, we have revised the corresponding section, with the modifications located in lines 333-339 and 470–485.

“The PCA analysis of the 19 climatic factors (Fig 8) showed that the first three principal components accounted for 87.21% of the original information, with PC1 (BIO3, BIO1, BIO12) explaining 60.89%, PC2 (BIO6, BIO11, BIO9) explaining 15.23%, and PC3 (BIO13, BIO16, BIO19) explaining 11.09%. These results indicated that PC1 and PC2 collectively captured the majority of climatic variability across sampling sites. To further explore the collective influence of climatic variables on LTN parameters, we performed correlation analysis between PCs(BIO3, BIO6, and BIO13) and LTN characteristics ” and “Climatic factors are the primary environmental drivers influencing network parameters such as average path length, diameter, edge density, and average clustering coefficient. Their explanatory power generally exceeds that of soil factors. The most significant impact was observed on the average clustering coefficient, explaining 63.47% of its variance. Although soil factors exhibit relatively smaller independent effects, they significantly influenced modularity (p<0.05), indicating their non-negligible role in shaping certain leaf trait network parameters—particularly when interacting with climate, which can alter network structure. Overall, soil factors significantly influenced specific parameters; however, climatic factors remained dominant, confirming the climate-driven nature of inland river basin ecosystems in arid regions. Soil indirectly influenced functional differentiation by regulating modularity (p<0.05), while the climate-soil interaction demonstrated a synergistic effect on edge density (p<0.05). This synergy may arise from coupled water-nutrient mechanisms, such as osmotic regulation of soil total potassium under drought stress, which modulates trait associations. The strong correlations between principal components (PCs) and leaf trait network (LTN) parameters underscore that climatic effects on Populus euphratica are not isolated but operate through synergistic interactions among multiple variables.”

5.The statistical analysis presented here in this study is weaker than even the authors’ first publication on the same dataset. For example, for statistics described (CV) in Figure 3 and 5, it is required to report at the least bootstrap standard error values. It is helpful for readers to know some measure of confidence in these summary stats especially given that there are only 20 sampling points in this study. Another example is that authors do not report the statistical significance of the variance component estimates described in Figure 10. One can use a method of permutation testing under full and reduced models using the same vegan R-package to do this with ease. There seems to be a general lack of disregard to statistical rigor shown throughout the manuscript. For example, claims made in lines 307-309 cannot be supported using statistical evidence. Therefore, it is strongly suggested that this is remedied in the revised submission.

In accordance with your suggestions, we have revised the corresponding section, with the modifications located in lines 239-244 and 473–480.

“Variance partitioning analysis (VPA) was performed using the "vegan" package in R to quantify the proportional contributions of climate, soil, and their interaction effects on network parameters. Permutation tests (999 permutations) were conducted via the permutest function to assess the statistical significance of these contributions (p-values) at a significance level of α=0.05. This approach was used to interpret the effects of different environmental factors (climate and soil) on the variability of network parameters.” and “Although soil factors exhibit relatively smaller independent effects, they significantly influenced modularity (p<0.05), indicating their non-negligible role in shaping certain leaf trait network parameters—particularly when interacting with climate, which can alter network structure. Overall, soil factors significantly influenced specific parameters; however, climatic factors remained dominant, confirming the climate-driven nature of inland river basin ecosystems in arid regions. Soil indirectly influenced functional differentiation by regulating modularity (p<0.05), while the climate-soil interaction demonstrated a synergistic effect on edge density (p<0.05)..”

6.Lines 66-76: This entire section contains repetitive words and ideas. It is not entirely clear what the authors are trying to convey in this section as the sentences are too long and convoluted.

In accordance with your suggestions, we have revised the corresponding section, with the modifications located in lines 67-72.

“These variations result from the interaction of environmental factors and phylogenetic history, showing dynamic changes in global, regional, and local distributions.”

7.Lines 104-106: Please cite the ‘few reports’ mentioned in this section. This is especially required since it sets up the premise for this manuscript. Mention how this investigation can be set apart from a very similar analysis carried out by Yao et al., 2024 cited in this manuscript since the methodology and the population in question are the same.

In accordance with your suggestions, we have revised the corresponding section.

Li Y, Liu C, Sack L, et al. Leaf trait network architecture shifts with species‐richness and climate across forests at continental scale[J]. Ecology Letters, 2022, 25(6): 1442-1457.

Yan J, He Y, Jiao M, et al. Leaf trait network variations with woody species diversity and habitat heterogeneity in degraded karst forests[J]. Ecological Indicators, 2024, 160: 111896.

8.Line 136: Should be DBH range 20-30 cm instead of 20>30 cm?

In accordance with your suggestions, we have revised the corresponding section, with the modifications located in lines 144.

“20-30 cm”

9.Line 155: This notation for range is unconventional. Numbers should be separated by a hyphen or as ranges. For example, [0, 20] or (0, 20]. First indicates that the zero and twenty are included and the second notation states that the range includes values greater than zero up to and including twenty.

In accordance with your suggestions, we have revised the corresponding section, with the modifications located in lines 163.

“(0, 20],(20,40], (40,60], (60,80], and (80,100] ”

10.Figure 3: It will be helpful for the readers if colors in the sub-figure (a) match that of (b). Although I presume that shades of green were used in this figure since they depict attributes of green leaves, I feel a more color-blind friendly scheme would be more appropriate here.

In accordance with your suggestions, we have revised the corresponding section, with the modifications located in lines 269-270.

1.Can you please upload an additional copy of your revised manuscript that does not contain any tracked changes or highlighting as your main article file. This will be used in the production process if your manuscript is accepted. Please amend the file type for the file showing your changes to Revised Manuscript w/tracked changes.

Thank you for pointing out the discrepancy in the clean copy. We have carefully reviewed and amended the clean copy to ensure consistency across all submission materials.

2.In your Methods section, please provide additional information regarding the permits you obtained for the work. Please ensure you have included the full name of the authority that approved the field site access and, if no permits were required, a brief statement explaining why.

Thank you for your suggestion. The field sampling of Populus euphratica in Xinjiang did not require specific permits, as per the regulations of the Xinjiang Uygur Autonomous Region Regulations on the Protection of Wild Plants and the People's Republic of China Forest Law. Populus euphratica is not classified as a nationally protected wild plant in China, and the study area is located in a non-protected public region along the Tarim River Basin, Xinjiang. The sampling activities involved only the collection of naturally shed branches and leaves from Populus euphratica and soil samples from the forest understory, without involving live tree sampling, protected species, or sensitive areas (e.g., nature reserves or ecological redline zones). We have strictly complied with the guidelines and regulations for field research activities issued by the Forestry and Grassland Bureau of Xinjiang Uygur Autonomous Region.

[The National Natural Science Foundation of China (Grant number 31160110)].

Thank you for your professional reminder. We hereby formally clarify the funding statement as follows: The funding agency of the National Natural Science Foundation of China (Grant number 31160110) had no role in study design, data collection and analysis, decision to publish, or preparation of the manuscript."

4.We note that Figure 1 in your submission contain [map/satellite] images which may be copyrighted. All PLOS content is published under the Creative Commons Attribution License (CC BY 4.0), which means that the manuscript, images, and Supporting Information files will be freely available online, and any third party is permitted to access, download, copy, distribute, and use these materials in any way, even commercially, with proper attribution. For these reasons, we cannot publish previously copyrighted maps or satellite images created using proprietary data, such as Google software (Google Maps, Street View, and Earth). For more information, see our copyright guidelines: http://journals.plos.org/plosone/s/licenses-and-copyright.

Thank you for bringing the copyright issue of Figure 1 to our attention. We have carefully addressed this concern by completely replacing the original map/satellite images with open-access data from Natural Earth (https://www.naturalearthdata.com/downloads/10m-raster-data/), which is freely available and licensed under a Creative Commons Attribution-4.0 International License (CC BY 4.0).

Data Replacement:

All proprietary or copyrighted elements (e.g., satellite imagery, topographic layers) were removed.

The map was redrawn using Natural Earth’s public domain vector and raster datasets (e.g., administrative boundaries, elevation, hydrography).

1.I deem all the reviewers' concerns have been addressed well in the revised manuscript. However, it seems the authors oversaw my own comments, one of which being "The authors recently published a paper using the same dataset, as noticed by Reviewer #2, and it is suggested to explicitly state that this manuscript is the statistical rework of the mentioned study." I do not see any mention on this issue in the manuscript, nor can I find any reply in response to reviewers. This is crucial information, and readers would need to be familiarized with this important fact.

Thank you for your suggestion. We add this suggested response to line 89 of the manuscript.

---

## [Editor Report · Decision Letter 2]

6 Apr 2025

Climate: The dominant factor influencing the spatial distribution pattern of the leaf trait network of Populus euphratica  along the main stream of the Tarim River

PONE-D-24-52929R2

Dear Dr. Huang,

We’re pleased to inform you that your manuscript has been judged scientifically suitable for publication and will be formally accepted for publication once it meets all outstanding technical requirements.

Kind regards,

Branislav T. Šiler, Ph.D.

Academic Editor

PLOS ONE
---

## [Editor Report · Acceptance letter]

PONE-D-24-52929R2

PLOS ONE

Dear Dr. Huang,

I'm pleased to inform you that your manuscript has been deemed suitable for publication in PLOS ONE. Congratulations! Your manuscript is now being handed over to our production team.

Kind regards,

on behalf of

Dr. Branislav T. Šiler

Academic Editor

PLOS ONE